# AgriGPT-VL: Agricultural Vision–Language Understanding Suite

## Abstract

Despite rapid advances in multimodal large language models, agricultural applications remain constrained by the scarcity of domain-tailored models, curated vision–language corpora, and rigorous evaluation. To address these challenges, we present the **AgriGPT-VL Suite**, a unified multimodal framework for agriculture. Our contributions are threefold. First, we introduce **Agri-3M-VL**, the largest vision–language corpus for agriculture to our knowledge, curated by a scalable multi-agent data generator; it comprises 1M image–caption pairs, 2M image-grounded VQA pairs, 50K expert-level VQA instances, and 15k GRPO reinforcement learning dataset. Second, we develop **AgriGPT-VL**, an agriculture-specialized vision–language model trained via a progressive curriculum of textual grounding, multimodal shallow/deep alignment, and GRPO refinement. This method achieves strong multimodal reasoning while preserving text-only capability. Third, we establish **AgriBench-VL-4K**, a compact yet challenging evaluation suite with open-ended and image-grounded questions, paired with a multi-metric evaluation and an LLM-as-a-judge framework. Experiments show that AgriGPT-VL outperforms leading general-purpose VLMs on AgriBench-VL-4K, achieving higher pairwise win rates in the LLM-as-a-judge evaluation. Meanwhile, it remains competitive on the text-only AgriBench-13K with no noticeable degradation of language ability. Ablation studies further confirm consistent gains from our alignment and GRPO refinement stages. All resources will be released to support reproducible research and deployment in low-resource agricultural settings at https://anonymous.4open.science/r/AgriGPT-VL-DA65/

## 1 Introduction

The convergence of AI with critical sectors like agriculture presents a significant opportunity to address global challenges such as food security and sustainable resource management (Swaminathan, 2001; Foley et al., 2011; Clapp, 2020). With the increasing challenges posed by climate change, resource scarcity, and population growth, intelligent agricultural decision-making is becoming indispensable (Godfray et al., 2010; Rockström et al., 2017; Fan & Rue, 2020). In recent years, multimodal large language models (MLLMs) have demonstrated remarkable progress in integrating vision and language, enabling tasks such as captioning, visual question answering (VQA), and multimodal reasoning (Yin et al., 2023; Achiam & et al., 2023; Chen & et al., 2024). While Multimodal Large Language Models (MLLMs) excel at integrating vision and language on general web data (Schuhmann et al., 2022; Du et al., 2022), they are ill-equipped for the agricultural domain. The knowledge required for tasks in crop and soil science is highly specialized and absent from standard pre-training corpora (Kamilaris & Prenafeta-Boldú, 2018; Wolfert et al., 2017). Consequently, existing MLLMs struggle with agricultural terminology, exhibit factual inaccuracies, and fail to provide reliable, context-aware support for real-world farming operations (Wu et al., 2024; Rezayi et al., 2022; Yang & et al., 2025).

Several attempts have been made to build agricultural language models, such as AgriBERT (Rezayi et al., 2022), AgriLLM (Didwania et al., 2024), AgroLLM (Samuel et al., 2025), and AgroGPT (Awais et al., 2025). These efforts show the value of domain-specific adaptation but are often constrained to text-only settings or narrow task coverage. Our earlier work, AgriGPT (Yang & et al., 2025), introduced the first agriculture-specialized LLM ecosystem with a curated instruction dataset

(Agri-342K), a retrieval-enhanced reasoning module (Tri-RAG), and a benchmark suite (AgriBench-13K). While effective for textual reasoning, AgriGPT lacked visual grounding and thus could not address multimodal agricultural tasks such as pest recognition or crop diagnosis. On the other hand, general-purpose MLLMs such as InternVL (Chen et al., 2024), Qwen-VL (Team, 2024), Gemini (Achiam & et al., 2023), and LLaVA (Liu et al., 2023b) demonstrate strong vision–language capabilities but are trained primarily on internet-scale data describing common objects, scenes, and events, which fail to capture agricultural semantics. As a result, these models suffer from hallucinations, poor transferability, and lack of reasoning ability in agriculture-specific scenarios. Related domains such as medicine developed specialized multimodal LLMs, highlighting the need for a comparable ecosystem in agriculture.

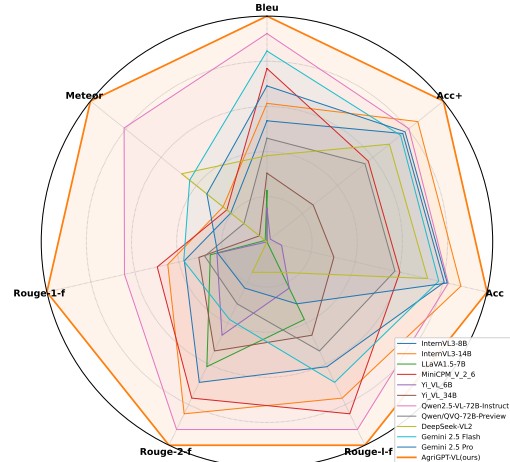

Figure 1: AgriGPT-VL achieves leading performance on AgriBench-VL-4K.

**Our contributions can be summarized as follows:**

- **Agri-3M-VL Dataset & Data Generator.** We build a transferable, reusable multi-agent Data Generator and use it to curate **Agri-3M-VL**: **1M** image–caption pairs, **2M** high-quality image-grounded VQA pairs, **50K** expert-level VQA, and **15k** rewarded GRPO reinforcement learning dataset(10K VQA + 5K single-choice questions). To the best of our knowledge, this is the largest agriculture vision–language corpus to date.

- **AgriGPT-VL & Curriculum Training.** Using a progressive curriculum, we train the agriculture-specialized VL model **AgriGPT-VL**, as shown in figure 1, which surpasses most flagship models in capability. To the best of our knowledge, this is currently the only open-source agriculture vision–language large model.

- **AgriBench-VL-4K & Evaluation Framework.** We construct a comprehensive and challenging benchmark with **2,018** open-ended VQA and **1,858** image-grounded single-choice questions (two per image for cross-consistency), coupled with multi-metric evaluation and a dual evaluation framework that includes an LLM-as-a-judge for pairwise preference.

- **Reproducible Resources.** We have open-sourced the datasets, models, and evaluation tools to support reproducible research and deployment in low-resource agricultural settings; URL: https://anonymous.4open.science/r/AgriGPT-VL-DA65/

## 2 RELATED WORK

### 2.1 TEXT-ONLY LANGUAGE MODELS IN AGRICULTURE

Pioneering work in agricultural AI largely focused on the language modality. Early models such as AgriBERT (Rezayi et al., 2022) adapted language model pre-training to domain-specific text corpora. Subsequent efforts, including AgriLLM (Didwania et al., 2024), AgroLLM (Samuel et al., 2025), and AgriGPT (Yang & et al., 2025), advanced this paradigm by developing large-scale instruction datasets like Agri-342K and text-only benchmarks such as AgriBench-13K (Yang & et al., 2025). For instance, Zhu et al. (Zhu et al., 2024) reviewed the progression of text-only and multi-modal agricultural LLMs, highlighting the transition from domain adaptation to instruction-based fine-tuning. Moreover, Yu and Lin (Yu & Lin, 2024) proposed a framework leveraging LLMs for agricultural knowledge inference and consultation, suggesting broader utility beyond QA. While these models demonstrated strong textual understanding, their primary limitation was the absence of visual grounding, restricting their applicability to tasks that do not require image interpretation.

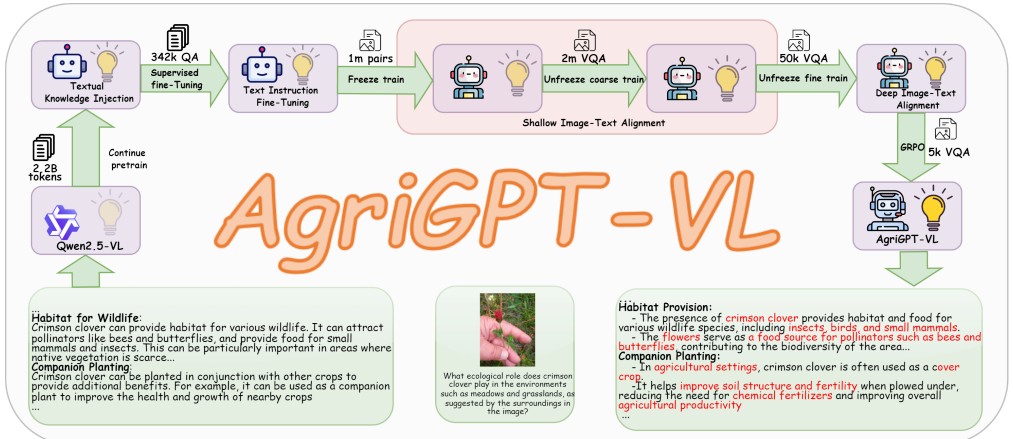

Figure 2: Overview of the AgriGPT-VL training pipeline and curriculum-based model evolution.

## 2.2 Emergence of Multimodal Agricultural Systems

The integration of visual data marked a critical evolution in agricultural AI. Foundational datasets like PlantVillage (Hughes & Salathé, 2015) and IP102 (Wu et al., 2019) provided large-scale image collections for specific recognition tasks, such as pest and disease identification. More recent works have begun to build multimodal models and benchmarks with broader capabilities. For instance, Agri-LLaVA (Wang et al., 2024), AgriCLIP (Nawaz et al., 2024), and LLMI-CDP (Wang et al., 2025) introduced vision–language abilities, while datasets like VL-PAW (Yu et al., 2025) and benchmarks like AgMMU (Gauba et al., 2025), AgroBench (Shinoda et al., 2025), and AgriEval (Yan et al., 2025) introduced tasks such as VQA and captioning. Other studies such as Zhu et al. (2024) provide a systematic review of the current landscape, while Yu & Lin (2024) and Arshad et al. (2025) explore concrete frameworks or empirical evaluations of VLMs in agricultural use cases. However, these multimodal resources often remain limited in scale, are restricted to narrow recognition tasks, or lack rigorous, large-scale quality control, representing disparate efforts rather than a cohesive foundation.

## 2.3 The Need for a Unified Vision–Language Ecosystem

The limitations of prior work highlight a clear need for a comprehensive and unified framework. While previous efforts have made valuable contributions to datasets, models, or benchmarks individually, progress has been hampered by the lack of a single ecosystem that integrates all three components at scale. To address this fragmentation, our work introduces a cohesive suite of resources. Our **AgriGPT-VL dataset** provides scale and quality; our **AgriGPT-VL** model handles complex reasoning beyond simple recognition; and our **AgriBench-VL-4K** benchmark enables robust, multifaceted evaluation. Together, these components form the kind of unified foundation we argue is necessary for the next generation of agricultural AI.

## 3 AgriGPT-VL

### 3.1 Dataset

Constructing training data is a fundamental challenge in developing multimodal large language models. To address this, we introduce the Data Generator, a transferable paradigm for systematically transforming raw images into high-quality multimodal instructions. The generator is designed not only for agriculture but also as a generalizable methodology that can be applied to other scientific domains where multimodal resources remain scarce or noisy.

As shown in Figure 4, we aggregated a wide range of datasets covering pests and diseases, insects, crops, weeds, and fruits. Specifically, the PlantVillage dataset contains 54,305 images across 38 classes (Abdallah, 2019). For insect-related data, we included 6,878 images covering 166 fine-grained insect species from the Species196 dataset (He et al., 2023), and the Insect Foundation dataset with 317,128 images spanning 38,867 fine-grained insect classes (Yu, 2020), totaling 324,006 images

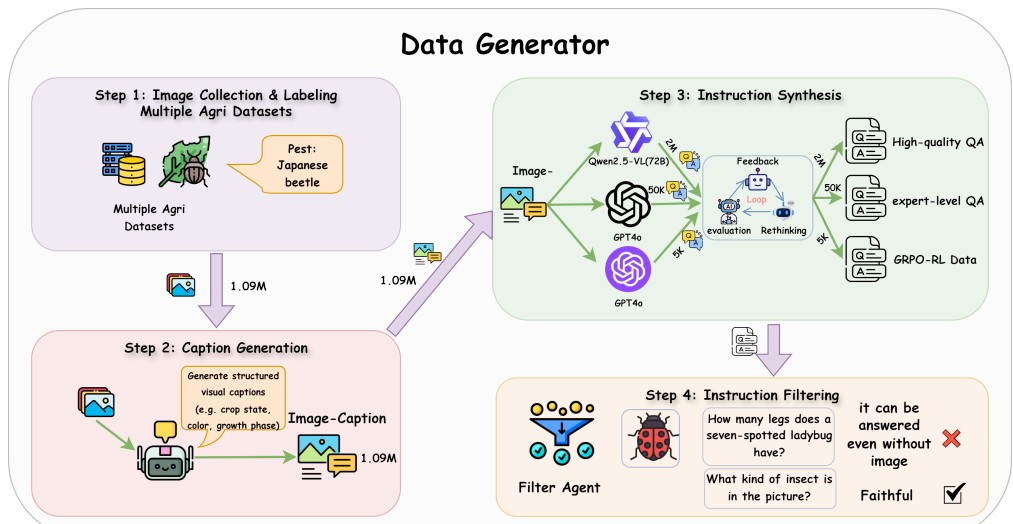

Figure 3: Data Generator: A multimodal instruction data generation pipeline.

and 39,033 classes. In the crop and weed domain, the SelectDataset provides 558,930 images over 2,958 categories (Contributors, 2021). For fruits, we incorporated Fruits-360 with 97,255 images and 206 categories (Murean & Oltean, 2018), and Fresh-Rotten Fruit with 30,357 images and 18 categories (Densu341, 2022), amounting to a combined 157,969 images and 224 classes. Altogether, these datasets cover **1,064,853** images and **42,253** fine-grained categories, nearly encompassing the full agricultural visual landscape.

However, these raw datasets suffer from several limitations: many lack descriptive annotations, exhibit inconsistent labeling, and cannot be directly used for multimodal model training. These shortcomings necessitate our proposed Data Generator, which systematically transforms such raw images into structured, instruction-ready corpora. Through several stages of processing, the Data Generator enables the creation of a large-scale, high-quality multimodal training corpus suitable for agricultural vision–language modeling.

As shown in figure 3, the Data Generator transforms multi-source agricultural images into instruction-ready corpora via four stages— caption generation, instruction synthesis, multi-agent refinement and instruction filtering yielding 1M image captions, 2M high-quality VQA, a 50K expert-level VQA, and 15k GRPO reinforcement learning dataset.The detailed high-quality VQA are illustrated in figure 5.

**(1) Caption Generation.** For the collected images spanning pests and diseases, insects, weeds, and fruits, we first generate structured visual captions. These captions describe observable attributes such as crop growth stage, leaf color, fruit maturity, or pest morphology. For example, an image of diseased tomato leaves is captioned with information about lesion color and spread, while

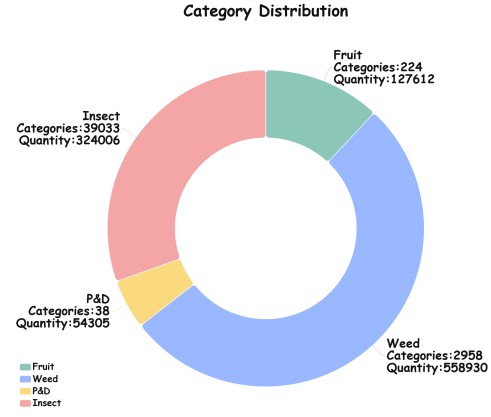

Figure 4: Category distribution of the dataset.

a fruit image records ripeness stage and external texture. In total, this stage yields about 1 million image–caption pairs, providing a descriptive foundation for subsequent instruction synthesis. The significance of this step is that captions transform raw visual data into semantically rich text, enabling downstream models to link domain-specific imagery with meaningful language.

**(2) Instruction Synthesis.** Building upon the image–caption pairs, we employ large vision–language models (e.g., Qwen2.5-VL 72B, GPT-4o) to generate diverse instructions and answers. This stage produces multiple types of VQA: high-quality factual queries, expert-level reasoning tasks, and in-

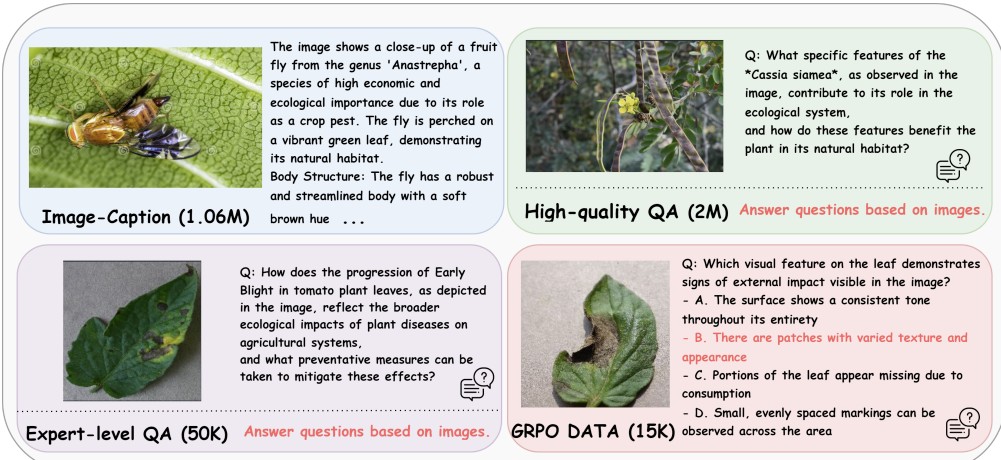

Figure 5: The four types of hierarchical training data constructed for AgriGPT-VL.

teractive multimodal dialogues. For instance, a weed image may lead to questions such as "What species of weed is shown?" (recognition) or "What is the likely impact of this weed on crop yield?" (reasoning). Altogether, we synthesize approximately 2 million VQA samples, covering both open-ended and single-choice formats. This step is essential because it elevates the dataset from simple recognition to instruction-following reasoning, directly aligning with the needs of multimodal LLMs.

(3) **Multi-Agent Refinement.** Based on the preceding Image–Caption data, we generate about two VQA pairs per image. Quality is controlled by a loop formed by three agents—Feedback, Evaluation, and Rethinking. Feedback proposes revisions, Evaluation scores samples along dimensions such as factual consistency and image grounding, and Rethinking rewrites with self-consistency checks; samples iterate among the three until preset thresholds are met, yielding roughly 2M high-quality VQA. Separately, we use GPT-4o to verify and polish an additional 50K subset for supervised fine-tuning, and we additionally construct 15k GRPO reinforcement learning dataset for reward modeling and preference optimization.

**(4) Instruction Filtering.** Finally, we introduce a filter agent to discard irrelevant or hallucinated instructions. For example, generic questions unrelated to the image (e.g., "How many legs does a seven-spotted ladybug have?") are removed, while faithful image-grounded questions (e.g., "What kind of insect is in the picture?") are retained. After filtering and manual verification, about 50K expert instructions remain, representing the most reliable and domain-specific supervision signals. This filtering step guarantees the factual alignment of data, mitigating hallucinations and improving trustworthiness in model training.

Each stage is complementary: caption generation provides semantic grounding, instruction synthesis injects reasoning diversity, multi-agent refinement structures feedback-driven selection, and instruction filtering enforces factual reliability. Together, they form a robust agricultural multimodal dataset that not only supports AgriGPT-VL training but also serves as a blueprint for dataset construction in other scientific domains.

## 3.2 MODEL

This section details our training paradigm for AgriGPT-VL, which follows a progressive curriculum: textual grounding first, then vision–language alignment. We first consolidate domain knowledge and instruction style on text-only data, and then align vision and language on synthesized multimodal supervision with an easy-to-hard schedule.

**Stage-1 (text-only domain grounding).** Starting from Qwen2.5-VL, we conduct continual pretraining on 200K documents (≈2.2B tokens) to inject agricultural terminology and background knowledge, followed by supervised instruction tuning on Agri-342K (Yang & et al., 2025). A held-out split of AgriBench-13K (Yang & et al., 2025) is used for early stopping and calibration prior to multimodal alignment.

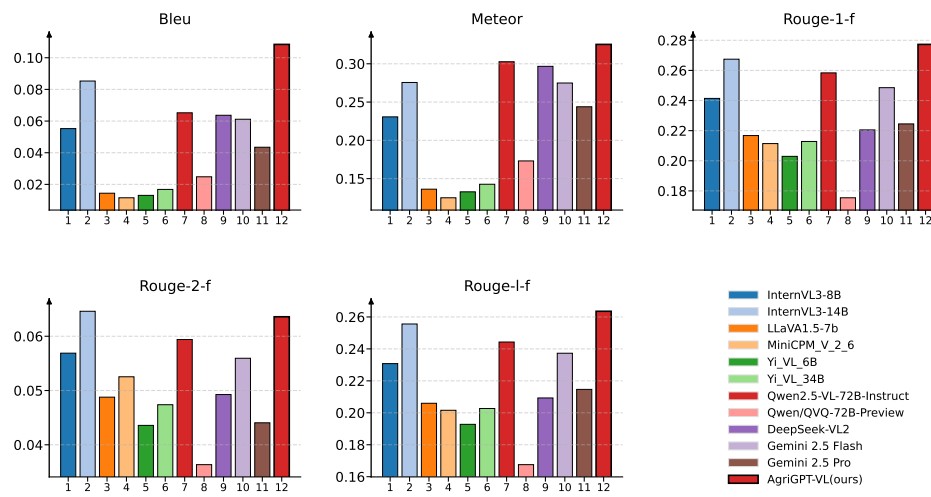

Figure 6: Text-only evaluation of vision-language models on AgriBench-13K.

**Stage-2 (curricular alignment on synthesized multimodal data).** We adopt a three-step easy-to-hard sequence built on caption and VQA supervision, then preference optimization:

**(2a) Shallow Alignment (Captioning, Vision Frozen).** We start with 1M image–caption pairs, keeping both the vision encoder and LLM component fully frozen. Only the connector and adapter layers are trained. Captioning tasks help establish a stable semantic bridge between vision and language modalities.

**(2b) Deep Alignment (From Coarse to Full Reasoning).** Next, we train on 2M image–QA samples (two questions per image for cross-validation), covering recognition, attributes, diagnosis, and basic multi-hop reasoning. Using LoRA, we gradually unfreeze the vision encoder and LLM, enabling transition to full multimodal reasoning.

**(2c) GRPO Optimization (Reward-Guided Fine-Tuning).** We build an expert pool via multi-agent refinement, selecting 50K GPT-4o-polished samples for supervised fine-tuning, and 15K GRPO samples for reinforcement learning. GRPO rewards image-text consistency, internal logic, and verifiable terminology. Details are in Appendix A.2 and A.6.

### 3.3 BENCHMARK-VL-4K

AGRIBENCH-VL-4K jointly evaluates generative and discriminative abilities with two components built from agricultural images: 2,018 open-ended question–answer pairs and 1,858 single-choice questions. Each image is paired with two single-choice questions to cross-validate predictions and reduce random guessing. The benchmark is strictly de-duplicated against all training splits and then human-reviewed to ensure fairness and reproducibility.

**Construction.** Open-ended questions are synthesized from structured captions of held-out images, covering recognition, symptom/mechanism analysis, management recommendations, and simple multi-step reasoning; answers are normalized for synonyms, units, and terminology. For the single-choice part, two complementary questions are generated per image. Distractors are mined from confusable taxa and co-occurring conditions to increase discriminability. To mitigate leakage, stems and option templates are designed not to overlap with training prompts.

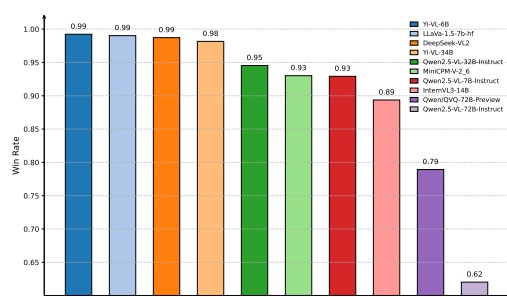

Figure 7: Pairwise win rate of vision-language models vs. AgriGPT-VL (LLM-judged)

**Quality control and de-duplication.** We remove near-duplicates at both the image and text levels: perceptual hashing or visual-feature simi-

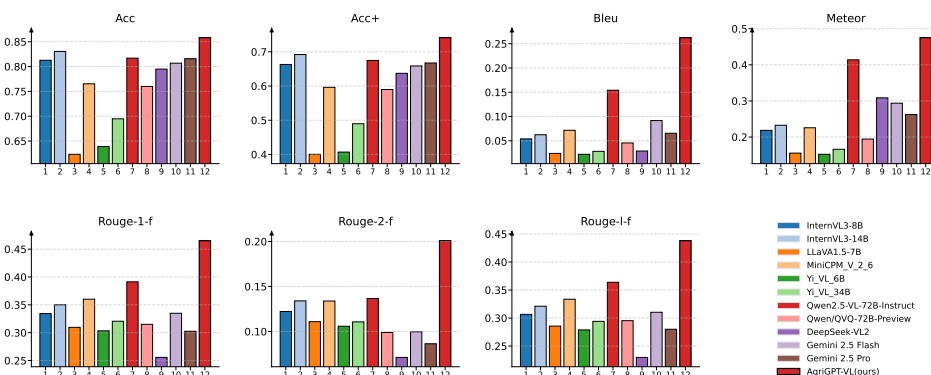

Figure 8: Multimodal evaluation of vision-language models on AgriBench-VL-4K.

Table 1: Language-only evaluation (text capability only). Comparison of AgriGPT-VL with general VLMs on text tasks without images. **Bold** and underlined denote best and second-best per column.

| Model | BLEU | Meteor | Rouge-1-f | Rouge-2-f | Rouge-L-f |
|---|---|---|---|---|---|
| InternVL-3-8B | 5.52 | 23.07 | 24.14 | 5.69 | 23.08 |
| InternVL-3-14B | 8.53 | 27.56 | 26.75 | **6.46** | 25.56 |
| LLaVA-1.5-7B | 1.44 | 13.62 | 21.67 | 4.88 | 20.60 |
| MiniCPM-V-2.6 | 1.15 | 12.50 | 21.41 | 5.25 | 20.16 |
| Yi-VL-6B | 1.03 | 12.38 | 20.93 | 4.36 | 20.09 |
| Yi-VL-34B | 1.69 | 14.27 | 21.82 | 4.74 | 20.27 |
| Qwen-VL-7B | 7.70 | 30.17 | 24.16 | 4.97 | 22.86 |
| Qwen2.5-VL-72B-Instruct | 6.52 | 30.27 | 25.84 | 5.95 | 24.43 |
| Qwen-QVQ | 2.48 | 17.31 | 17.54 | 3.63 | 16.75 |
| DeepSeek-VL-1.2 | 6.37 | 29.67 | 22.10 | 4.93 | 20.93 |
| Gemini-2.5-Flash | 6.12 | 27.49 | 24.85 | 5.59 | 23.73 |
| Gemini-2.5-Pro | 4.34 | 23.69 | 24.16 | 4.45 | 22.30 |
| AgriGPT-VL (ours) | **10.84** | **32.53** | **27.73** | 6.36 | **26.36** |

larity for images, and lexical/embedding similarity for question–answer strings. De-duplication is applied across train–evaluation as well as within the evaluation split. All remaining items undergo a two-pass human review by agriculture-literate annotators focusing on factual correctness, image-grounded evidence, and ambiguity resolution, with adjudication for disagreements.

## 4 RESULTS

### 4.1 COMPARATIVE EXPERIMENT

We focus on two questions: (i) after progressively injecting domain knowledge and vision–language alignment, is textual competence preserved and strengthened; and (ii) in real image–language settings, does the model exhibit stronger visual grounding and agronomic reasoning—i.e., can it both choose correctly (discriminative robustness) and articulate evidence–based answers (generation quality). To this end, we evaluate text–only capability on AgriBench-13K (Yang & et al., 2025)and multimodal capability on AgriBench-VL-4K. For discriminative evaluation, we report *Acc* (single–choice accuracy, scored per question) and $Acc^+$ (image–level cross–consistency: both single–choice questions for the same image must be correct). For generation, we report BLEU (Papineni et al., 2002), METEOR (Banerjee & Lavie, 2005), and ROUGE-L (Lin, 2004) to measure terminology conformity, semantic coverage, and structural completeness.

We compare *AgriGPT-VL* against twelve representative vision–language models: InternVL-3-8B/3-14B (Zhu et al., 2025), LLaVA-1.5-7B (Liu et al., 2023a), MiniCPM-V-2.6 (Yao et al., 2024; OpenBMB Team, 2024), Yi-VL-6B and Yi-VL-34B (01.AI, 2024b;a), Qwen-VL-7B (Bai et al., 2023), Qwen2.5-VL-72B-Instruct (Bai et al., 2025), Qwen-QVQ (Qwen Team, 2024), DeepSeek-VL-1.2 (Lu et al., 2024), and Gemini-2.5 Flash/Pro (Google DeepMind & Google AI, 2025a;b).

Table 2: Vision–language evaluation (multimodal capability). Comparison of AgriGPT-VL with general VLMs on image-grounded tasks.

| Model | Acc | Acc$^+$ | BLEU | Meteor | Rouge-1-f | Rouge-2-f | Rouge-L-f |
|---|---|---|---|---|---|---|---|
| InternVL-3-8B | 81.27% | 66.31% | 5.38 | 21.85 | 33.43 | 12.22 | 30.69 |
| InternVL-3-14B | 83.05% | 69.21% | 6.32 | 23.26 | 35.01 | 13.44 | 32.11 |
| LLaVA-1.5-7B | 62.33% | 40.04% | 2.39 | 15.55 | 30.96 | 11.09 | 28.57 |
| MiniCPM-V-2.6 | 76.53% | 59.63% | 7.12 | 22.57 | 36.02 | 13.39 | 33.36 |
| Yi-VL-6B | 63.89% | 40.69% | 2.21 | 15.21 | 30.32 | 10.59 | 27.88 |
| Yi-VL-34B | 69.48% | 48.98% | 2.83 | 16.61 | 32.05 | 10.98 | 29.42 |
| Qwen2.5-VL-72B-Instruct | 81.70% | 67.49% | 15.41 | 41.38 | 39.14 | 13.25 | 36.63 |
| Qwen/QVQ-72B-Preview | 76.00% | 58.99% | 4.55 | 19.43 | 31.46 | 9.09 | 29.53 |
| DeepSeek-VL2 | 79.49% | 63.72% | 2.88 | 30.86 | 25.57 | 7.21 | 22.97 |
| Gemini-2.5-Flash | 80.68% | 65.88% | 9.12 | 29.38 | 33.47 | 10.00 | 31.33 |
| Gemini-2.5-Pro | 81.59% | 66.74% | 6.55 | 26.22 | 30.25 | 8.65 | 28.01 |
| AgriGPT-VL (ours) | **85.84%** | **74.17%** | **26.27** | **47.55** | **46.52** | **20.09** | **43.81** |

Table 3: Ablation study of alignment stages. **Bold** indicate best per column.

| Setting | Acc | Acc$^+$ | BLEU | Meteor | Rouge-1-f | Rouge-2-f | Rouge-L-f |
|---|---|---|---|---|---|---|---|
| Base(Qwen2.5-VL-7B) | 77.20% | 60.32% | 13.42 | 38.24 | 35.52 | 10.78 | 32.73 |
| + Shallow Alignment | 78.23% | 62.47% | 15.54 | 36.47 | 40.76 | 14.07 | 37.95 |
| + Shallow + Deep | 81.18% | 66.67% | 21.68 | 44.38 | 43.04 | 15.62 | 40.37 |
| + Shallow + Deep + GRPO | **85.84%** | **74.17%** | **26.27** | **47.55** | **46.52** | **20.09** | **43.81** |

As shown in tables 1 and figure 6, on AgriBench-13K (Yang & et al., 2025), AgriGPT-VL leads across mainstream text metrics, indicating that the progressive training does not sacrifice language ability; instead, it strengthens standardized use of agricultural terminology and canonical answer style, consolidating textual representations and providing a stable linguistic base for the subsequent multimodal stage.

As shown in table 2 and figure 8, on AGRIBENCH-VL-4K, we obtain the best results on all metrics, surpassing several flagship large models. Gains in *Acc* reflect more precise image–option matching; gains in *Acc*$^+$ demonstrate consistent semantics per image and stronger resistance to hard distractors (confusable taxa and co–occurring conditions), thereby mitigating chance guessing and better reflecting true capability. Improvements in Bleu (Papineni et al., 2002), Meteor (Banerjee & Lavie, 2005), and Rouge-1-f/Rouge-2-f/Rouge-L-f (Lin, 2004) further indicate three strengthened abilities: (1) visual evidence grounding and factor extraction (organs, colors/lesions, phenology); (2) agronomic multi-step reasoning (from symptoms to plausible causes and management consistent with scene constraints); and (3) professional, audit-ready expression (units, terminology, and thresholds that follow domain conventions).Detailed definitions and computation formulas of the evaluation metrics are included in Appendix A.5

In addition, as shown in figure 7, we conduct JudgeLM (Jiao et al., 2023) blind pairwise comparisons: for each query, two systems' outputs are judged head-to-head, we swap left/right positions to reduce order bias, and average the two outcomes. We report three preference metrics: *WR* (ties excluded). Across most strong baselines, AgriGPT-VL achieves consistently higher win rates and remains competitive against top large models, corroborating the above advantages from a preference perspective. Appendix A.3 describes the prompt design for the LLM-based judger, and Appendix A.4 details the metric computation methodology.

## 4.2 ABLATION STUDY

Starting from a base model, we progressively add *Shallow Alignment* (caption-only supervision with the vision stack frozen to establish cross-modal semantic anchors), *Deep Alignment* (single-choice reasoning with the vision encoder and cross-modal interaction layers unfrozen), and *GRPO* (reinforcement optimization with 15k GRPO reinforcement learning dataset).

As shown in tables 3 and figure 9, the results reveal a clear hierarchy of contributions: Shallow Alignment primarily improves lexical and descriptive consistency, stabilizing image–text keypoint

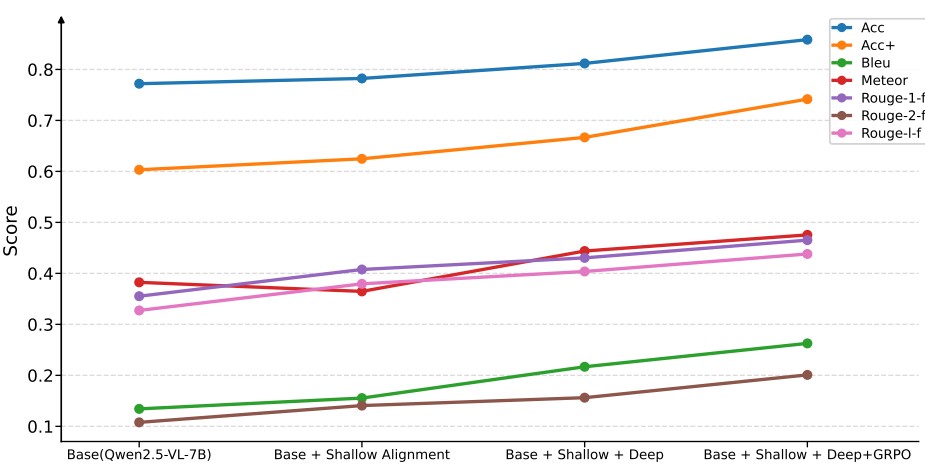

Figure 9: Ablation study on curriculum-based alignment stages of AgriGPT-VL

Table 4: Evaluation of general capabilities before and after fine-tuning. Numbers in the second row indicate dataset sample sizes.

| Model | MMLU | ARC | OpenBookQA | MMBench | MMMU | SeedBench |
|---|---|---|---|---|---|---|
| Samples | 5701 | 7787 | 5958 | 10698 | 1035 | 17023 |
| Qwen2.5-VL | 0.6783 | 0.9043 | 0.8501 | 0.8398 | 0.4329 | 0.7565 |
| AgriGPT-VL | 0.6741 | 0.8462 | 0.8412 | 0.8312 | 0.4599 | 0.7574 |

correspondence; Deep Alignment is the main driver of cross-modal understanding and reasoning, lifting both discriminative and generation metrics; and GRPO further enhances factual faithfulness and robustness, with the largest gains on the stricter image-level cross-consistency metric ($Acc^+$), indicating that expert-level instructions are necessary to constrain high-precision behavior.

### 4.3 GENERALIZATION EVALUATION

To assess whether domain specialization preserves general capabilities, we compare the fine-tuned *AgriGPT-VL* with its base model (*Qwen2.5-VL*) on six public benchmarks: three text-only (MMLU (Hendrycks et al., 2021), ARC (Clark et al., 2018), OPENBOOKQA (Mihaylov et al., 2018)) and three vision–language (MMBENCH (Liu et al., 2025), MMMU (Yue et al., 2024), SEED-BENCH (Li et al., 2024)).

Overall, *AgriGPT-VL* remains competitive. On text-only tasks, performance is largely preserved on MMLU and OPENBOOKQA, with only a modest decline on ARC. On vision–language tasks, the model matches or exceeds the base, showing parity on SEED-BENCH and MMBENCH, and clear gains on MMMU.

As shown in Table 4, two conclusions emerge: (i) the curriculum—starting with textual grounding—effectively mitigates forgetting, maintaining broad competence across ~40K out-of-domain samples; (ii) the improvement on MMMU confirms that learned visual reasoning generalizes beyond agriculture, reinforcing the strength and transferability of our finetuning framework.

## 5 CONCLUSION

We present AgriGPT-VL, an agricultural vision–language understanding suite that unifies large-scale data generation, curriculum-based multimodal training, and benchmark evaluation. The model demonstrates strong agronomic reasoning and visual grounding without sacrificing general capabilities. This compact and reproducible framework provides a practical blueprint for building specialized multimodal systems in agriculture and beyond.

## 6 REPRODUCIBILITY STATEMENT

We take multiple steps to ensure the reproducibility of our work, covering three major components: the construction of hierarchical multimodal data, the curriculum-aligned training of AgriGPT-VL, and comprehensive benchmarking in agricultural vision–language tasks.

For **data construction**, Section 3.1 presents the data sources and statistical distributions (see Figure 4), the construction pipeline is illustrated in Figure 3, and representative examples of the four types of hierarchical data (captioning, QA, expert QA, and GRPO-based preference data) are shown in Figure 5. We release all data generation scripts, annotation formats, and documentation to support reproduction and reuse.

For **model training**, the curriculum-based alignment strategy is described in Section 3. All hyper-parameter configurations across different stages are detailed in Appendix A.2, including batch size, learning rate, LoRA rank ($r$), LoRA scaling factor ($\alpha$), dropout rate, number of training epochs, device type, and GPU count.

For **evaluation**, the benchmark protocol is introduced in Section 4. The evaluation metrics, including single-choice accuracy (Acc), image-level strict accuracy (Acc$^+$), BLEU, METEOR, ROUGE-1/2/L, are formally defined in Appendix A.3. We provide all evaluation scripts and baseline configurations.

All datasets, models (pretrained and fine-tuned), training scripts, evaluation tools, and benchmark task files have been fully released at **https://anonymous.4open.science/r/AgriGPT-VL-DA65/** (anonymous link). This release enables the community to reproduce our results, verify the performance, and build upon our hierarchical agricultural VL foundation.

## 7 ETHICS STATEMENT

This work focuses on developing an open agricultural vision–language model, AgriGPT-VL, to support downstream applications such as intelligent farming assistance, crop health monitoring, and agricultural education. All datasets used or constructed in this study are composed of publicly available or synthetically generated images, and do not contain personal information, biometric data, or sensitive demographic attributes.

The entire training and evaluation pipeline is conducted in a simulated setting without involvement of human subjects or private data. We take care to avoid using copyrighted or restricted datasets.

From a broader perspective, AgriGPT-VL is intended to benefit global agriculture, particularly in resource-constrained regions where AI-driven assistance may improve food production, reduce manual labor, and promote sustainable practices. However, as with any foundation model, misuse is possible. To mitigate risks, we openly share our datasets, training methodology, and evaluation results, and encourage transparent and responsible usage within the research community.

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

# A APPENDIX

## A.1 GENAI USAGE DISCLOSURE

Large Language Models (LLMs) were utilized in this work for minor writing refinement, such as typo correction and linguistic polishing. Given that our research directly investigates LLMs, they were also employed in data construction and evaluation processes. All uses of generative models were carefully controlled and fully compliant with academic integrity standards. No generative tools were used for code generation, figure/table creation, or experimental manipulation beyond the scope described above.

## A.2 EXPERIMENTAL SETTINGS

We summarize the detailed hyperparameter configurations for all fine-tuning strategies used in our experiments. These strategies include continued pretraining, supervised instruction tuning, full-parameter training, staged unfreezing, and reinforcement learning–based methods such as GRPO. LoRA is used in most setups, with consistent dropout and adaptation parameters for fair comparison.

Table 5: Detailed hyperparameter settings for all fine-tuning methods. Most methods adopt LoRA-based parameter-efficient tuning. Freeze and full tuning use different configurations as baselines.

| Method | Tuning | Batch | Device | LR | LoRA $r$ | LoRA $\alpha$ | Dropout | Epoch |
|---|---|---|---|---|---|---|---|---|
| Continue pretrain | LoRA | 8 | 8×RTX 4090 | 5e–8 | 8 | 16 | 0.05 | 1 |
| Supervised fine-tune | LoRA | 8 | 8×RTX 4090 | 1e–7 | 8 | 16 | 0.05 | 1 |
| Freeze train | Full | 16 | 8×RTX 4090 | 1e–7 | – | – | – | 1 |
| Unfreeze coarse train | LoRA | 16 | 8×RTX 4090 | 1e–8 | 8 | 32 | 0.05 | 1 |
| Unfreeze fine train | LoRA | 16 | 8×RTX 4090 | 2e–4 | 8 | 32 | 0.05 | 1 |
| GRPO-1 | LoRA | 8 | 8×RTX 4090 | 5e–5 | 8 | 32 | 0.05 | 1 |
| GRPO-2 | LoRA | 8 | 8×RTX 4090 | 5e–6 | 8 | 32 | 0.05 | 1 |

## A.3 PROMPT FORMATTING DETAILS FOR DATA CONSTRUCTION AND EVALUATION

We detail below the prompt templates used for judgment, image captioning, QA generation, filtering, and single-choice question creation. These were used to construct and evaluate our dataset and models.

---

**Judgement Prompt**

You are a helpful and precise assistant for checking the quality of the answer. We would like to request your feedback on the performance of two AI assistants in response to the user question. Please rate the helpfulness, relevance, accuracy, and level of details of their responses. Each assistant receives an overall score on a scale of 1 to 10, where a higher score indicates better overall performance.

Please first output a single line containing only two values indicating the scores for Assistant 1 and 2, respectively. The two scores are separated by a space.

In the subsequent line, please provide a comprehensive explanation of your evaluation, avoiding any potential bias and ensuring that the order in which the responses were presented does not affect your judgment.

**[Question]** {q} **[Reference Answer]** {reference}

**[The Start of Assistant 1's Answer]** {answer_1} **[The End of Assistant 1's Answer]**

**[The Start of Assistant 2's Answer]** {answer_2} **[The End of Assistant 2's Answer]**

---

**Image Caption Generation Prompt**

You will be provided with an image and related information about the image. Your task is to describe the image in as much detail as possible, making full use of the given information. Focus on providing a clear, accurate, and comprehensive description.

**[Image Related Information]** {category}

---

**QA Generation Prompt**

You will be given an image and a short description of that image. Your task is to design 2 insightful questions about the image and provide comprehensive answers to each.
**Requirements:**

- Ground your questions and answers strictly in the image and its description; avoid unsupported speculation.

- Be specific about visual evidence (objects, attributes, actions, spatial relations, text in the image, colors, composition, context).

- Each question must start with the label `<question>` and end with `</question>`, and each answer must start with the label `<answer>` and end with `</answer>`.

**[Image Description]** {image_description}
**[Output Format]** `<question>` {question} `</question>` `<answer>` {answer} `</answer>`

---

**Filter Prompt (Image–Text Relevance)**

You are an expert image–text alignment evaluator. You will be given a question and an image. Your task is to determine whether the given question is relevant to the content of the image. In other words:

- If answering the question requires examining the image, respond with `Relevant`.

- If the question can be answered without reference to the image, respond with `Irrelevant`.

**[Input]** Question: {question} Image: {image}
**[Output Format]** `Relevant` or `Irrelevant`

---

**single-choice Question Generation Prompt**

You are an expert in agriculture and dataset creation. Given the following image description, generate **one multiple choice question** that relies **strictly on visual evidence from the image**, not on general knowledge.
**[Image Description]** {description}
Here are some example items for reference:
```
<question> What feature in the image suggests that the infection has
not reached an advanced stage?
A. Entire leaf is curled and brown
B. Lesions have concentric rings and pycnidia
C. Most of the leaf surface is still green and intact
D. The veins are completely degraded </question>
<answer> C </answer>
<question> Where is the largest necrotic lesion located on the leaf?
A. Near the central midrib
B. On the left lower margin
C. At the leaf tip, expanding inward
D. Along the stem attachment </question>
<answer> C </answer>
```
**Requirements:**

- Must be based on observable content in the image.

- Include four answer choices (A–D), with only one correct answer.

- Each question must start with `<question>` and end with `</question>`.

- Each answer must start with `<answer>` and end with `</answer>`.

## A.4 Judger Metric Calculation

To eliminate position bias in pairwise comparisons, we conduct each evaluation twice: in the second round, the order of the two candidate responses is swapped. The final score is computed as the average of the two rounds.

This symmetric evaluation ensures fairness and robustness by reducing the influence of response position on annotator judgments.

---

**WR2: Pure Win Rate (excluding ties)**

This metric excludes ties and focuses on the proportion of wins among non-tie cases, reflecting absolute superiority.

$$\text{WR} = \frac{\#win}{\#all - \#tie} \tag{1}$$

It captures the model's ability to dominate when a clear winner is determined.

---

## A.5 Metric Definitions for Multimodal Evaluation

We adopt a combination of single-choice accuracy metrics and text generation metrics to evaluate model performance on different task types. Definitions are provided below.

---

**Acc: Answer Accuracy**

**Acc** measures the proportion of correctly answered single-choice questions over all questions.

$$\text{Acc} = \frac{\#correct\_answers}{\#total\_questions} \tag{2}$$

Each question is judged independently.

---

**Acc$^+$: Image-level Strict Accuracy**

**Acc$^+$** evaluates whether both questions associated with the same image are answered correctly.

$$\text{Acc}^+ = \frac{\#images\_with\_both\_questions\_correct}{\#total\_images} \tag{3}$$

This stricter metric captures the consistency of understanding per image.

---

**BLEU**

BLEU (Bilingual Evaluation Understudy) is a precision-based metric that computes the n-gram overlap between the generated and reference text.

$$\text{Bleu} = \exp\left(\min\left(1 - \frac{r}{c}, 0\right) + \sum_{n=1}^{N} w_n \log p_n\right) \tag{4}$$

Where $r$ is reference length, $c$ is candidate length, and $p_n$ is the modified n-gram precision.

---

**METEOR**

METEOR computes unigram precision and recall, considering stemming and synonymy, and penalizes fragmented matches.

$$\text{Meteor} = F_{\text{mean}} \times (1 - Penalty) \tag{5}$$

Where $F_{\text{mean}}$ is the harmonic mean of precision and recall, and Penalty is based on chunk fragmentation.

---

**ROUGE-1/2/L-f**

ROUGE measures recall-based n-gram and longest common subsequence overlap. We report ROUGE-1, ROUGE-2, and ROUGE-L with F1 scores:

- **Rouge-1-f**: Unigram overlap F1 score
- **Rouge-2-f**: Bigram overlap F1 score
- **Rouge-L-f**: Longest common subsequence-based F1

---

### A.6 GROUP RELATIVE POLICY OPTIMIZATION (GRPO)

#### A.6.1 GRPO OBJECTIVE

During GRPO training, for each iteration and a given input $q$, we sample $M$ candidate outputs from the previous policy $\pi_{\text{ref}}$. Each candidate $j$ receives a reward $r_j$, and we compute the *group-relative advantage* as

$$\tilde{A}_j = \frac{r_j - \nu}{\tau}, \qquad \nu = \frac{1}{M}\sum_{j=1}^{M} r_j, \quad \tau = \sqrt{\frac{1}{M}\sum_{j=1}^{M}(r_j - \nu)^2}, \tag{6}$$

where $\nu$ and $\tau$ denote the mean and standard deviation of rewards within the candidate group. The clipped surrogate objective of GRPO is

$$\mathcal{L}_{\text{GRPO}}(\phi) = \mathbb{E}_{y_j \sim \pi_\phi}\left[\frac{1}{M}\sum_{j=1}^{M}\min\left(\varrho_j \tilde{A}_j, \ \text{clip}(\varrho_j, 1-\epsilon, 1+\epsilon)\,\tilde{A}_j\right)\right] - \lambda\,\text{KL}\left[\pi_\phi \,\|\, \pi_{\text{ref}}\right], \tag{7}$$

where $\pi_\phi$ is the updated policy, $\varrho_j = \frac{\pi_\phi(y_j|q)}{\pi_{\text{ref}}(y_j|q)}$ is the importance-sampling ratio, $\epsilon$ is the clipping hyperparameter, and the KL penalty with weight $\lambda$ constrains policy deviation.

#### A.6.2 EXACT-MATCH REWARD

We first define a binary reward that checks whether the predicted answer matches the reference exactly. Let $\text{parse}_{\text{sol}}(\cdot)$ extract the ground-truth answer from $||\ldots||$ delimiters if present (otherwise the raw string), and $\text{parse}_{\text{out}}(\cdot)$ extract the model's answer from $||\ldots||$ delimiters if present. After normalization $\text{norm}(\cdot)$ (e.g., whitespace trimming), we define

$$g_j = \text{norm}\big(\text{parse}_{\text{sol}}(\text{solution}_j)\big), \qquad \hat{y}_j = \text{norm}\big(\text{parse}_{\text{out}}(\text{output}_j)\big). \tag{8}$$

The reward is then

$$r_j^{\text{exact}} = \begin{cases} 2.0, & \text{if } \hat{y}_j = g_j, \\ 0.0, & \text{otherwise}, \end{cases} \tag{9}$$

and substituting $r_j^{\text{exact}}$ into Eq. equation 6 yields exact-match advantages $\tilde{A}_j$.

#### A.6.3 SEMANTIC REWARD

To provide a more graded supervision signal, we also introduce a semantic reward based on similarity metrics. Given a solution $g$ and a model output $\hat{y}$, we compute:

- BLEU$(\hat{y}, g)$ using $n$-gram overlap,
- METEOR$(\hat{y}, g)$ considering stemming and synonyms,
- ROUGE-L$_F(\hat{y}, g)$ measuring longest common subsequence overlap.

We then combine them as

$$r_j^{\text{sem}} \; = \; \frac{\text{BLEU}(\hat{y}_j, g_j)}{0.16} \; + \; \frac{\text{METEOR}(\hat{y}_j, g_j)}{0.40} \; + \; \frac{\text{ROUGE-L}_F(\hat{y}_j, g_j)}{0.30}, \tag{10}$$

where the denominators are scaling constants ensuring comparable ranges. This yields a continuous-valued reward, encouraging the model to align more closely with reference answers even when partially correct.

