# OpenReview forum: "AgriGPT-VL: Agricultural Vision–Language Understanding Suite"
_ICLR.cc/2026/Conference — ICLR 2026 Conference Withdrawn Submission_

### Official Review · Reviewer_92zj · 2025-10-27

**Soundness:** 3
**Presentation:** 2
**Contribution:** 2
**Rating:** 6
**Confidence:** 3

**Summary:**

This paper introduces AgriGPT-VL, a comprehensive agricultural vision-language system comprising three components: (1) Agri-3M-VL, a large-scale multimodal dataset with 1M image-caption pairs, 2M VQA pairs, 50K expert-level VQA, and 15K GRPO samples; (2) a curriculum-trained vision-language model achieving state-of-the-art performance on agricultural benchmarks; and (3) AgriBench-VL-4K, an evaluation suite with 2,018 open-ended and 1,858 single-choice questions. The authors claim this is the first open-source agricultural VLM, demonstrating superior performance over general-purpose models including much larger systems (72B parameters).

**Strengths:**

1. The work provides a complete pipeline, data generation framework, training methodology, and evaluation benchmark.
2. The multi-agent Data Generator (Figure 3) with iterative refinement loops offers a transferable methodology applicable to other scientific domains with limited multimodal resources.
3. AgriGPT-VL achieves consistent improvements over 12 baselines across multiple metrics (Table 2: 85.84% Acc, 74.17% Acc+), including commercial systems like Gemini 2.5 Pro.
3. The benchmark incorporates cross-consistency checks (Acc+ metric), multiple text generation metrics, and LLM-as-judge pairwise comparisons to reduce evaluation bias.
4. Table 4 demonstrates that domain specialization maintains competitive performance on general benchmarks (MMLU, MMBench), with improvements on MMMU suggesting transferable visual reasoning.

**Weaknesses:**

1. Circular reasoning and potential data contamination (Lines 213-215, Table 2)

The 7B model outperforms its 72B teacher (Qwen2.5-VL-72B) used for data generation, raising two concerns:

 - Does AgriBench-VL-4K test genuine agricultural understanding or memorization of synthetic patterns from the Qwen family?
 - Generation metrics (BLEU: 26.27 vs. 15.41) may favor the student's learned distribution over actual quality.

Could you provide (a) perplexity analysis showing the benchmark is not biased toward Qwen-generated text, (b) human evaluation comparing AgriGPT-VL vs. Qwen2.5-VL-72B on the same questions?

2. Insufficient quality control documentation (Lines 355-357)
The paper mentions "two-pass human review" but lacks critical details:

- Did reviewers verify factual correctness, image grounding, answer quality, or all three?
- Were reviewers agriculture-trained, or general crowdworkers?
- No inter-annotator agreement (Cohen's κ, Fleiss' κ) or sample rejection rates reported.
- How were annotator performance and consistency monitored?

Could you add an appendix section detailing the annotation guidelines, annotator qualifications (N reviewers, background), agreement scores, and examples of accepted/rejected samples.

3. Missing error analysis of synthetic data (Section 3.1)

- What percentage of generated captions/QA contain factual errors (e.g., incorrect pest identification, wrong disease symptoms)?
- How often do models hallucinate visual details not present in images?
- Are there domain-inappropriate responses (e.g., temperate-climate advice for tropical crops)?

My suggestion: sample 500 instances, manually audit for error types, and report error rates per category.

4. Incomplete GRPO methodology (Lines 242-243, Appendix A.6)
The 15K GRPO dataset construction is underspecified: How were these 15K samples chosen from the 2M pool? Random sampling, hardest negatives, or diversity-based selection?


5. No failure case analysis (Section 4)

- Performance breakdown by task type: pest identification vs. disease diagnosis vs. management recommendations?
- Difficulty analysis: early-stage vs. late-stage disease detection accuracy?
- Crop specificity: does the model generalize across different crops, or is performance concentrated on common species?
- Geographic/climate robustness: performance on region-specific agricultural practices?

Add a subsection analyzing failure modes with concrete examples and error taxonomy.

6. Figure 2 is hard. I would suggest to put some examples in it and reduce the model name a bit. It's taking too much space.
7. Citation formatting (Line 161): "Yu, 2020" textual citations instead of linkable one.

**Questions:**

1. Which specific model generated the 1M captions? Was it Qwen2.5-VL-72B, GPT-4o, or another system?

2. Have you tested the model with actual farmers or agricultural extension workers?

3. Can you add the parameter count (7B) to all result tables for easier comparison with similarly-sized baselines?

---

### Official Review · Reviewer_5Qfd · 2025-10-30

**Soundness:** 3
**Presentation:** 3
**Contribution:** 2
**Rating:** 4
**Confidence:** 4

**Summary:**

This paper introduces the AgriGPT-VL Suite, a comprehensive framework for vision-language understanding in the agricultural domain. The authors make three primary contributions: 1) Agri-3M-VL, a large-scale agricultural vision-language dataset containing over 3 million instances (captions, VQA, and reinforcement learning data), created using a novel multi-agent data generation pipeline. 2) AgriGPT-VL, a specialized vision-language model trained on this data using a progressive curriculum that includes textual grounding, multimodal alignment, and reinforcement learning. 3) AgriBench-VL-4K, a new benchmark for evaluating agricultural VLMs, complete with a multi-metric and LLM-as-a-judge evaluation framework.

**Strengths:**

1. The authors don't just release a model or a dataset, but an entire ecosystem (data, generator, model, benchmark). This is a substantial effort that lowers the barrier to entry for other researchers in this area.
2. The paper includes a wide range of strong baseline models, a convincing ablation study that validates their training curriculum, and a generalization study to ensure the model doesn't suffer from catastrophic forgetting.
3. The paper is exceptionally well-written and organized, making a complex set of contributions easy to understand.

**Weaknesses:**

1. The "Data Generator" is presented as a key contribution but is effectively a black box. The description lacks the necessary detail for reproducibility. What are the specific prompts, models, or algorithms that constitute the "Feedback," "Evaluation," and "Rethinking" agents?
2. The model is evaluated on a benchmark created by the same pipeline used for its training data. This calls into question the validity of the large performance gaps observed against other models. The work would be much stronger if evaluated on other independent benchmarks, such as AgriBench[1], AgMMU[2] and SeedBench[3]
3. The paper appears to violate the double-blind review policy. On lines 52-53, the authors explicitly reference "Our earlier work, AgriGPT (Yang & et al., 2025)" a citation that directly reveals their identities. Furthermore, the claim of introducing the "first" agriculture-specialized LLM is an overstatement, as other contemporary or prior works, such as SeedLLM [4], also exist.

[1] Yutong Zhou and Masahiro Ryo. AgriBench: A Hierarchical Agriculture Benchmark for Multimodal Large Language Models. arXiv preprint arXiv:2412.00465, 2024.

[2] Aruna Gauba, Irene Pi, Yunze Man, Ziqi Pang, Vikram S Adve, and Yu-Xiong Wang. Agmmu: A comprehensive agricultural multimodal understanding and reasoning benchmark. arXiv preprint arXiv:2504.10568, 2025.

[3] Ying, Jie, et al. "SeedBench: A Multi-task Benchmark for Evaluating Large Language Models in Seed Science." arXiv preprint arXiv:2505.13220 (2025).

[4] Yang F, Kong H, Ying J, et al. SeedLLM· Rice: A large language model integrated with rice biological knowledge graph[J]. Molecular Plant, 2025, 18(7): 1118-1129.

**Questions:**

1. Could you please provide a more detailed elaboration of the "Data Generator"? Specifically, the descriptions of the "Feedback," "Evaluation," and "Rethinking" agents are unclear.
2. Can you address the concern regarding the evaluation benchmark being "in-distribution" with the training data's generation style? How can we be sure that the performance gains are due to agronomic reasoning and not simply overfitting to the generator's style? Have you considered evaluating AgriGPT-VL on other independent agricultural benchmark (e.g., AgriBench[1], AgMMU[2] and SeedBench[3]) to validate its capabilities in a more neutral setting?
3. On lines 52-53, you refer to "Our earlier work, AgriGPT (Yang & et al., 2025)". This appears to be a direct violation of the double-blind review policy. Could you explain this serious issue? You claim that this work introduces "the first" agricultural LLM, but I believe this is an overclaim, as an earlier model, SeedLLM [4], already exists.

[1] Yutong Zhou and Masahiro Ryo. AgriBench: A Hierarchical Agriculture Benchmark for Multimodal Large Language Models. arXiv preprint arXiv:2412.00465, 2024.

[2] Aruna Gauba, Irene Pi, Yunze Man, Ziqi Pang, Vikram S Adve, and Yu-Xiong Wang. Agmmu: A comprehensive agricultural multimodal understanding and reasoning benchmark. arXiv preprint arXiv:2504.10568, 2025.

[3] Ying, Jie, et al. "SeedBench: A Multi-task Benchmark for Evaluating Large Language Models in Seed Science." arXiv preprint arXiv:2505.13220 (2025).

[4] Yang F, Kong H, Ying J, et al. SeedLLM· Rice: A large language model integrated with rice biological knowledge graph[J]. Molecular Plant, 2025, 18(7): 1118-1129.

---

### Official Review · Reviewer_KjMx · 2025-10-31

**Soundness:** 2
**Presentation:** 2
**Contribution:** 2
**Rating:** 2
**Confidence:** 3

**Summary:**

The paper presents AgriGPT-VL, a unified vision–language understanding suite for agriculture.
It introduces three key components: 1) Agri-3M-VL, a large-scale agricultural vision–language dataset (1M image–caption pairs, 2M VQA pairs, 50K expert-level QA, and 15K GRPO samples) built via a multi-agent data generator. 2) AgriGPT-VL model, trained through a curriculum of textual grounding, shallow and deep multimodal alignment, and GRPO reinforcement optimization on top of Qwen2.5-VL. 3) AgriBench-VL-4K, an evaluation benchmark with 2,018 open-ended and 1,858 single-choice image-grounded questions, combined with multiple metrics and an LLM-as-a-judge framework. Experiments show that AgriGPT-VL outperforms several general-purpose VLMs (e.g., Gemini, Qwen, InternVL) on agricultural tasks, while maintaining text-only performance. The work emphasizes reproducibility with released data, models, and scripts.

**Strengths:**

1. Comprehensive ecosystem: integrates data, model, and benchmark—rare for agriculture.
2. Reproducibility commitment: data and code are promised for release, hyperparameters reported.
3. Practical impact: could benefit low-resource agricultural or scientific domains.

**Weaknesses:**

1.	Methodological originality is low.
The pipeline largely repackages existing paradigms (instruction tuning + LoRA + RLHF/GRPO) without new theoretical insight or training objective.
2.	Opaque multi-agent refinement.
The Feedback/Evaluation/Rethinking and Filter agents appear to be role-prompted LLMs, but their architectures are not specified.
No quantitative evaluation of filtering accuracy or hallucination reduction is provided.
3.	Ambiguity of GRPO dataset construction.
The 15 k “reward modeling” dataset is described but not exemplified—unclear what the reward labels look like.
4.	Terminology drift.
“Visual grounding” is repeatedly used in a broad sense (image-grounded reasoning) rather than localization/region grounding; terminology should be revised for precision.
5.	Redundant figures/tables waste space and obscure key findings, e.g., Table 1 & Fig. 6, Table 2 & Fig. 8, Table 3 & Fig. 9.
6.	Engineering focus over conceptual insight.
The contribution is largely empirical; it would benefit from a more principled study of domain alignment or grounding metrics.
7.	Experimental fairness uncertain: Unclear if baseline models were fine-tuned on the same data or evaluated zero-shot.

**Questions:**

1.	Multi-Agent Refinement: 1) Are the Feedback, Evaluation, and Rethinking agents simply prompt-engineered GPT-4o/Qwen2.5-VL models? 2) What criteria or quantitative thresholds determine “quality” convergence? 3) Could you report inter-agent agreement or filtering performance for hallucination removal?
2.	Filter Agent: 1) How does it detect hallucination beyond image–text irrelevance? 2) Is there a verification step with human annotators or cross-model voting?
3.	GRPO Dataset: 1) Please show several examples. 2) Are rewards human-provided or computed automatically via metrics (BLEU/METEOR/ROUGE)?
4.	Terminology: Clarify the use of “visual grounding”. If region-level grounding is not used, consider renaming to “image-grounded reasoning” or “image–text consistency”.
5.	Baseline Training & Evaluation: 1) Were baselines fine-tuned on Agri-3M-VL or evaluated zero-shot? If fine-tuned, under identical compute and data budgets? 2) Please specify this clearly in the experimental setup and table captions.
6.	Presentation: 1) Consolidate redundant tables/figures to improve readability. 2) Include one qualitative case of the multi-agent refinement loop and its improvement effect.

---

### Note · Authors · 2025-11-14

I have read and agree with the venue's withdrawal policy on behalf of myself and my co-authors.